# Computer-Aided Greenery Design—Prototype Green Structure Improving Human Health in Urban Ecosystem

**DOI:** 10.3390/ijerph20021198

**Published:** 2023-01-10

**Authors:** Dominik Sędzicki, Jan Cudzik, Lucyna Nyka

**Affiliations:** Department of Urban Architecture and Waterscapes, Faculty of Architecture, Gdańsk University of Technology, 11/12 Narutowicza Street, 80-233 Gdansk, Poland

**Keywords:** greenery, automated design, sustainability, public health, landscape design, architecture

## Abstract

Increasing population and urbanization, with climate change consequences, such as rising temperatures, influence public health and well-being. The search to improve the quality of life in cities becomes one of the priority objectives. A solution can be found in the role of greenery in an urban environment and its impact on human health. This opens a path toward experimentation on microclimate green structures that can be inserted into dense urban spaces providing human and environmental benefits. The article proposes an automated greenery design method combined with rapid prototyping for such interventions. A theoretical analysis of the problem preceded the introduction of the method. The research process was developed in accordance with the main objectives of the CDIO framework (Conceive, Design, Implement, and Operate) with the SiL (Software in the Loop) and HiL (Hardware in the Loop) methods. Moreover, the applied test model allows for complex evaluation in order to ensure quality and directions for further development.

## 1. Introduction

The urbanization and rise of the population are leading to climate change with all its consequences; for example, increased pollution and rising temperatures that affect human health conditions [1]. Therefore, the search to improve the quality of life becomes one of the emerging issues. One of the topics that gained recent scientific interest is the role of greenery in an urban environment and its impact on public health. Greening of the cities is discussed against the problems of air pollution and urban ventilation [2,3], cooling effects for buildings [4], improving their energy balance, and mitigating the heat island effects in cities by reduction of air and surface temperatures [5,6,7]. Urban vegetation enhances biodiversity and provides various ecosystem services [8,9]. Access to green spaces has proven to positively affect human health by relieving stress and improving the well-being of citizens [10,11,12]. Contact with nature supports our psychological and mental health and fosters social ties [13,14]. Along with personal and social benefits, human health contributes to economic growth and achieving sustainable development goals (SDG) [15].

These findings give the direction of research inquiries and architectural practices experimentation toward enhancing relationships with nature in cities and developing new typologies of designing with greenery such as ecological urbanism, biophilic urbanism, or landscape urbanism [16,17,18]. Urban forests and parks are considered key elements of the urban green infrastructure that deliver multiple health services; thus access to green spaces should inform urban policies and planning decisions [19]. A large group of studies relates to the methods of incorporating vegetation into building envelopes in the form of living walls, green facades, and green roofs [20,21]. Their impact on human health has been widely discussed—green walls may reduce air pollution and noise, mitigate urban heat island effects, and regulate humidity [22]. However, as some research studies reveal, the realization of such solutions could be time-consuming and often encounters problems related to ownership, maintenance, or organization of the process with many stakeholders [23]. In many cases, the introduction of green roofs and other permanent architectural solutions demands major building retrofits.

This opens a path toward experimentation on easy-to-fabricate microclimate green structures that can be inserted into dense urban spaces in a relatively short time to provide human and environmental benefits. This field of experimentation has grown considerably in recent years with the advancements in digital technology and fast prototyping methods and tools [24]. However, as previous studies reveal, the integration of greenery into such structures may pose problems and be time-consuming since the decision on the size and distribution of holders for plants, selection of plants, and the shape of the structure takes place at the same time and the final result is developed in many iterative steps [25]. In effect, greenery is often introduced in the last stage of the design process and planted on the ground or from ground-mounted containers, and not integrated into the structure [26]. Thus, there is a need to develop scenarios, where vegetation could be introduced in a relatively short time, in accordance with chosen criteria, and fully integrated into the built form. Filling this gap, the paper presents a process of developing a prototype green structure with BIM (Building Information Modelling) tools, rapid fabrication, and the proposed method of automated greenery selection. The automated greenery design method is presented, tested, and discussed for possible further applications.

Implementing BIM technology to achieve health benefits in urban environments has been intensively explored concerning the dispersion of air pollutants, thermal radiation of public spaces, or noise and soundscape modeling [27]. BIM technology allows the management of information on the distribution of green areas, the density and patterns of vertical gardens, or the number and size of trees [28]. This can be used as a database for modeling and evaluating different aspects of urban structure and its impact on human health, such as access to green spaces from urban locations, the role of greenery in urban ventilation, or to predict deposition of chemical particles or allergenic pollen [29]. The 3D BIM model data was used to apply the AIROT (Aerobiological Index to create Risk maps for Ornamental Trees) index and allowed to map results as a graphical representation in the environmental sustainability study. This experiment’s results permit the visualization and understanding of the conditions of the studied area and tree species in the model, helping designers understand the role and constraints of the planned urban green infrastructure [30]. It has been applied in a study to minimize aerobiological risks in future new buildings or even for maintenance tasks of urban green infrastructure.

At the same time, the advances in digital technology and rapid prototyping tools allow for the fast fabrication of experimental urban structures that can be placed in the urban environment. Sustainability measures redirected hard approaches to buildings and urban structures toward soft (more organic and inspired by nature). The recent series of experimental building demonstrators developed by the Institute for Computational Design (ICD) and the Institute for Building Structures and Structural Design (ITKE) at the University of Stuttgart are bio-based and bio-inspired. One of them is a lattice-composite shell structure woven into one piece by an advanced robotic fabrication system [31]. The urge toward developing fast fabrication installations as future organic components of the urban structure is represented by another ICD/ITKE project, the Maison Fibre [32].

While such investigations are focused mainly on exploring boundaries between structural engineering, design, new materials, and fabrication constraints, some experimental urban initiatives and interventions use digital tools to design prototype constructions that support the growth of plants. Urban Microclimate Canopy—a Living Architecture Prototype, developed by the Faculty of Architecture, Technical University of Munich research team—is an example of a digitally fabricated fiberglass structure supporting climbing plants constructed to explore new ways of integrating vegetation in densely built urban environments. However, the process of plant selection was done without digital supporting tools. Moreover, for two short-term exhibitions, ivy plants were only temporarily installed in the structure [33].

While technological advancements facilitate the integration of new green fast fabricated prototypes into an urban structure, the idea of small temporary gardens in the city is not new. The inquiry into such forms of urban greenery intensified in the last decades with the emergence of environmental and health problems related to rapid urbanization processes. The environmental and landscape attributes, health benefits, and design challenges related to non-permanent landscapes have made them a creative and stimulating testing ground for researchers and designers. Raffaella Sini examines the historical evolution of the genre, exploring theories and strategies informing temporary small gardens. Key topics refer to temporary gardens as opportunities to work with live processes, and to gardening practice as a form of therapy and inclusion, which corresponds with social justice, public health, biodiversity, and ecology. Temporary gardens enrich the urban greenery forms and direct research studies toward a full range of new opportunities [34]. The recent findings on the health benefits of horticultural therapy, resulting not only from gardening but even viewing small gardens, particularly in the context of the aging society, reveal the need for working out innovative design solutions [35,36]. As Sini indicates, the time-limited design of small gardens affects the entire process of conceiving, building, experiencing, and managing green spaces; using short-term formats, anyone can invent, try, and experiment in a condensed experience of landscape [34].

In this context, the rapid prototyping methods and automatization of plant selection may bring new opportunities to the whole process. However, it is to be noted that automatization of the design process in the architectural field is a relatively new concept and only a few studies have taken up this topic [26]. Even though available software allows the use of visual algorithms or written scripts for the purpose of automatization processes, such methods are still not common in architecture and urban planning. Whereas, in the process of looking for the right answer to sustainability and health problems, in architecture, landscape architecture, and urban design, progress in pushing the boundaries is fundamentally associated with experimentation. Developing adequate research-by-design approaches is vital to provide architects, landscape architects, and urban designers with the knowledge and skills to meet the challenges of demographic, social, economic, environmental, and technological changes.

Thus, the purpose of this research is to propose a new, innovative method to develop easy-to-fabricate green structures that can be inserted into dense urban environments and thus contribute to public health and environmental benefits. The proposed approach is based on linking rapid prototyping with BIM tools and automated greenery selection. The presented study develops a new complex digital method of selection and design of greenery based on a parameter spreadsheet and environmental data. Parametrization of the chosen group of plants presented in this paper provides an initial step toward creating a database of plant species that can be used for algorithmic plant selection and distribution in further studies.

## 2. Materials and Methods

### 2.1. Methods

The research aimed to create a strategy for the development of a prototype microclimate installation based on green structure, rapid fabrication, and the method of automated greenery selection. The process of creating the test model for automated greenery design was based on elective research-by-design seminars organized for young researchers at the Faculty of Architecture at the Gdańsk University of Technology. The general process was conducted following the CDIO agenda and developed from its principles—SiL (software in the loop) and HiL (hardware in the loop) method developed for architectural research-by-design by Lucyna Nyka and Jan Cudzik [37].

The research was divided into three consecutive phases, starting with predesign, design, and ending with the fabrication and evaluation phase (Figure 1). The strategy applied in the predesign research started with the digitalization of the site which was supported by environmental analysis of temperature range, soil type, moisture, and sun exposure. The analysis led to the determination of the scope of the design task and parameters of plant species, materials, and technology. The basis for the automated greenery selection process was the algorithm that used the RhinoCommon Library to develop results based on the combination of multiple parameters with the use of a visual editor, Grasshopper^®^ (Robert McNeel & Associates, Seattle, WA, USA).

The digital analysis was supported by Ladybug^®^, a plugin for Rhinoceros^®^. The process led to necessary adjustments of automated greenery design (AGD) methods for the purpose of the presented research study [26]. The AGD method takes into account parameters such as sun exposure, temperature range, moisture, maintenance, soil type, and soil reaction. The analysis of these parameters allowed for the automated selection of specific plants according to the location and characteristics of the created architectural form.

The design phase consists of four consecutive steps. The first one was the design process carried out by the involved researchers. The second was the revisions and selection of the design for further development. The selected design was then revised with a computer algorithm automatically creating greenery design scenarios that were adopted and applied in the experiment. The last step of the design phase was the optimization of digital fabrication. In the phase of fabrication and evaluation, the first step was the application of a digital production that was later followed by manual processing and securing the material with impregnation. This step allowed for on-site construction. In the last stage of the process, the design was evaluated.

### 2.2. Materials

#### 2.2.1. Sample Scenario

The selection of location was based on general parameters: accessibility, good exposure to the sun, green surroundings, and safety. The chosen area is a part of the Gdańsk University of Technology campus, surrounded by sports facilities, the Pavilion of Architecture, and Steffens Park. The location is set between the railway line and 6 alley road (Figure 2).

The digital environmental analysis was conducted with Rhinoceros^®^ and Grasshopper^®^ supported with the Ladybug^®^ plugin [38]. The source for environmental data was an EnergyPlus Weather file (.epw) located in the Nowy Port district that was within acceptable distance for such analysis [39]. This analysis consisted of sun hours throughout the year and a range of temperatures [40]. Moreover, on-site soil analysis with the humidity and soil reaction was conducted. Due to the plan of putting specific plants in pots, a suitable type of soil was selected according to plant needs.

#### 2.2.2. Species Selection

For the purpose of the experiment, a group of climber plants was selected. These plants clothe walls and support foliage and flowers. Climbers cling by using tendrils, twining stems, stem roots, or sticky pads, while wall shrubs need to be tied to supports. Climbers mainly grow upwards and prefer sturdy supports to help them on their way [41]. This group of plants offers a range of sun-sensitivity from sun-loving to shade-loving, require relatively low maintenance, and are fast-growing species so that the effects of growth are noticeable even after one season [42]. Immersive spread of these plants invites planting participants to revisit the site of the experiment for further study. Because of these characteristics, climbers provide a group of plants that are well-fitted for a performance experiment in an architectural environment [43].

Many aspects of climbing plants have attracted scientific attention, from attachment mechanisms and general anatomy to genetic makeup and chemical properties [44]. Climbing vegetation growth has gained considerable research interest, from Darvin’s studies [45] to modern-day experimentations [46]. This group of plants is a good fit when designing for better well-being in cities. They are applicable in regard to particular health and environmental benefits such as removing air pollution [47] or improving the cooling performance of buildings [48].

The research on planting preferences and characteristics of growth for particular species provided parameters that were used for the Automated Greenery Design method (AGD). The sources of the data were several research studies regarding vegetation [49,50,51,52,53,54]; in particular, the characteristics of plant growth were based on “The Plant Growth Planner” by Caroline Boisset [53] and preferred sun exposure was developed according to “Begrünte Architektur” by Rudi Baumann [54]. The final data is the outcome of the evaluation and selection of available plant parameters. The research aimed to analyze and provide the set of parameters that affect plant vegetation in relation to the architectural environment. The data used in the research was created by the Laboratory of Digital Technologies and Materials of the Future, Faculty of Architecture, Gdańsk University of Technology. Parameterization of plants’ data conducted in the laboratory is part of a broader research program on the possibility of automation of design processes. An example of the gathered and curated data is presented in the table below (Table 1).

The group of plant species from the climbers family was chosen from an available database produced by the Laboratory of Digital Technologies and Materials of the Future. The most relevant criterion for this experiment was to provide a diversified spectrum of solar exposure so that any area on and around the planned installation can be matched with a particular species for planting. The chosen species are:*Clematis**Fallopia**Hedera helix**Parthenocissus*

In subsequent steps of the study, plants’ parameters in combination with climatic data allowed for the automatic analysis process to be carried out as an integrated element of spatial structure design.

#### 2.2.3. Experimental Model Construction: Material and Fabrication Technology Selection

For the strategy of building and testing the prototype to work, adequate consideration of construction material was necessary. The main drivers behind the choice of materials for the construction of a prototype green structure were determined by the project’s aim to implement the presence of greenery support in urban scenarios with minimum effort and maximum adjustment to the existing environment.

The other criteria for selecting the material were driven by level of complexity in terms of digital fabrication, accessibility, resilience to weather conditions, structural strength, ease of manual material processing, and low carbon footprint [55]. The considered materials were concrete, high-density Styrofoam, wood, plywood, carbon fiber, steel, and aluminum. After careful consideration plywood was selected. Plywood’s structural strength makes it an excellent choice for constructing installations and CNC production which allows the designer almost an infinite array of options [56]. Plywood parts can be joined in various ways and the technological aspects of joinery can be easily learned on a basic level. The material is produced in many factories worldwide and therefore provides access worldwide [57]. The material should be adjusted to the production location according to the type of trees grown locally. The environmental parameters were highly important in the selection process in order to embed the potential for raising eco-awareness into the structure itself [58].

The selection of digital fabrication methods considered laser cutting, advanced robotics, 3D printing, and CNC. The complexity of the use of robotics, limitations of scale in 3D printing, and impossibility to cut thick structural parts in wood with the use of laser cutting, all excluded these options for the production method. On the other hand, accessibility, ease of use, ability to cut structural parts in plywood, and low-energy post-production manual process all established the use of Computer Numerical Control machines as an option of choice [59]. CNC production allows plywood parts to be cut and routed with extreme accuracy, opening up a whole range of design possibilities for the design process. CNC cutting is currently being utilized by architects to make a whole range of different objects ranging from furniture, kitchens, built-in joinery, and prefabricated housing [60].

There are many different types of plywood available on today’s market, ranging from sheets formed in slow-grown dense tropical hardwoods to fast-grown lightweight softwoods. Since the installation was intended for an outdoor site, due to its ability to balance temperature and humidity variations as well as provide sufficient constructional properties, we have selected water-resistant birch plywood 18 mm thick and additional impregnation with environmentally neutral products.

## 3. Results

### 3.1. Design Process

The strategy of integrating young researchers into research and development tasks provided numerous solutions for the intended realization of the microclimate structure. (Figure 3). Priority of project selection was outlined in the design brief and included the project’s qualitative merits including plant distribution preferences, constructional innovation, buildability, and benefits to a surrounding environment. Instructions also called for a project that should demonstrate consideration of aesthetic, technical, functional, economic, ecological, and sustainability requirements to be encompassed in the design. The most successful design proposition was selected for further development and fabrication phase.

Architectural design is a process that integrates several different disciplines though some of them are not well represented in early-stages including greenery design. That often becomes only a part of the final design stages whereas it should be a part of all stages and therefore affect the decision-making process. The design agenda aimed mainly at solving this issue. The core idea was to allow for experimentation in search of the most versatile structures providing diversified conditions for selected species of plants. The test group consisted of 15 young researchers that were divided into six teams. Each team had the same goal, which was to design a form that would provide support and conditions for four types of climbing plants with different needs. Each group started with sketches which got reviewed and then worked on both digital and physical models which allowed for a better understanding of the form and better evaluation of the design process according to CDIO principia. The final presentation of proposals led to a discussion about achieved results and possible changes.

All of the projects provided acceptable solutions to design problems. The concept selected for the construction, however, distinguished itself from the others with the quality of achieved design goals. Firstly, the chosen proposal presented the most diverse distribution of greenery positions on the structure relative to different levels of sun exposure. This allowed for the automatic plant selection algorithm to use every plant species from the provided list when matching the designated spots with different kinds of vegetation. Secondly, the chosen design, when compared to the other proposals, offered the most efficient material usage. It used only 5 sheets of plywood, whereas most of the other designs used exceedingly more. Thirdly, the selected structure was the easiest to build. It used the smallest number of timber joints out of all proposals and also, in contrast with the other designs, provided specifically designed holes for the pots, therefore ensuring the easiest construction and fast greenery planting.

### 3.2. The Automated Selection Process

The automated greenery design (AGD) is a method for including plants in early-stage design processes, developed at the Digital Technologies and Materials of the Future Laboratory of the Gdańsk University of Technology. The AGD (Figure 4) allows for automated plant selection based on plant species parameters combined with site environmental analysis. Parameters that are considered can be adjusted according to the designers’ needs and characteristics of the project. In the presented experimental model, the following parameters were taken into consideration: sun exposure, temperature range, moisture, maintenance, soil type, and soil reaction.

To provide the parameters for the algorithm, a database was developed. The database has two forms of representation. The first one consisted of a series of parameters in the form of a spreadsheet in .csv file format that was applied to the algorithm. The second one was developed in the form of a table for presenting the parameters to the designer and user (Table 1). The environmental evaluation was based on on-site and digital analysis. The soil type and reaction were measured at the selected location. The humidity range, temperature range, and sun exposure (Figure 5) were calculated using digital analysis on environmental data from the EnergyPlus Weather file (.epw) located in the Nowy Port district.

The automation was conducted with Rhinoceros^®^, Grasshopper^®^, and Ladybug^®^ tools. The outcome of the process was the optimal scenario for plant selection computed for the chosen design of the structure (Figure 6). In the computational process, the environmental analysis was combined with plant parameters that allowed for the solution to be presented in the form of a schematic view that points out a specific plant location. Subsequently, this scenario was applied in a prototype experimental model.

### 3.3. Fabrication

After selecting the final form, a 3D model of the microclimate structure was recreated with the use of the McNeel Rhinoceros^®^ software (Version 6; Robert McNeel & Associates, Seattle, WA, USA) according to digital fabrication needs. The evaluation was then carried out to ensure buildability and optimization. The 3D model was revised to accommodate tolerances into joints in plywood and an additional cutting path was added to exclude the rounding of the tool bit out of the join slots (Figure 7) [61].

The next stage consisted of flattening parts of the installation and nesting these elements onto 1525 × 1525 mm plywood sheets to minimize material waste (Figure 8) [62]. All the parts were numbered and placed on sheets according to the building schedule to allow construction to start while other parts were being cut on the CNC plotter, therefore minimizing the time of construction. The aim of such an approach was to examine the possible solution for further development of model fabrication management.

When all the boards of plywood were prepared for cutting, the CNC movements had to be planned in the form of a g-code. The tool path was generated using RhinoCam^®^ software (Version 1.0; MecSoft Corporation, Dana Point, CA, USA), which is a plugin for the Rhinoceros^®^ (Figure 9). G-code was exported from the software in the form of a text file, consisting of a list of coordinates, and then imported into the CNC plotter [63]. The tool bit of choice was an up-down drill, 8 mm in diameter, which is used for fast and deep cutting action with minimal post-production needs.

While CNC milling was in progress the manual processing of already-cut parts began (Figure 10). The elements of the structure had to be sanded and then a coat of varnish was applied for additional water protection. As soon as the parts dried, they were joined with the rest of the prepared elements (Figure 11).

The process continued until the assembly was completed. The process of fabrication was completed within one working day according to schedule, without any prior preparation, and without occurring errors. Construction was assembled on-site (Figure 12). Moreover, throughout the time of process, all vegetation was planted in pots according to design decisions and to automatic plant selection protocol. The pots were filled with a layer of gravelite mixed with sand and then with a layer of soil. The whole mix was enriched with a composition of ecological fertilizers containing nitrogen and phosphorus compounds. The chosen climbers were planted in the pots and their elements were tied to the structure to ensure proper support.

### 3.4. Final Installation—Maintenance, Inspections, and Verification of the Method

The created prototype is an effect of analysis and searches for the most suitable design solution for the green structure that can be built in a relatively short time. The aim was not to automate the entire form-finding process but to connect the traditional design approach (manual design or BIM design) with the automated greenery design system. This allows for evaluating the effectiveness of the automated vegetation selection method and testing it on a controlled laboratory scale. The outcome is promising and shows the applicability and scalability of the method for more complex projects.

The design process of the created structure was set according to the CDIO model [37] with the application of SiL- and HiL-verification loops. The inspections allowed for a better understanding of growing patterns and possible risks that accompany them. The experimental prototype becomes evidence of the durability of selected materials and plants. Thus, it can become an important starting point for later experiments planned for a larger scale and other locations. With the evaluation of methods and extension of the plants’ library, there is an added value to the development of the researched field of knowledge.

Regarding the possibility of creating the installation, it was assumed that it should not exceed a working week. The design process was set accordingly to enable digital fabrication. The design was aiming for the maximum reduction of unnecessary connecting elements and waste. The above criteria are assessed positively, the form was designed, analyzed, and produced within 5 days, and the entire fabrication was carried out using a CNC machine. Manual processing was only a minor percentage of the whole task and could be reduced in the future.

As testing through the seasons (spring, summer, and fall) showed, the planted vegetation has grown according to the described growth pattern, and the conditions allowed for the diversification of plants in terms of light exposure (Figure 13). The placement of particular plants was set correctly according to the environmental needs with all plants completing the full vegetation circle. The prototype needed additional maintenance during the unexpectedly dry season which proves the need for site inspection.

Application of the method to the creation process of the experimental prototype made it possible to verify the effectiveness and efficiency of the vegetation selection method at the early stages of designing and to verify the parameterization of features. An important element of the supervision and observation process is also the possibility to draw conclusions regarding any additional parameters that should be considered in the process of parameterization of greenery features for the purposes of automation of design processes.

## 4. Discussion

According to the presented research method, temporary or seasonal installations can be designed and later developed into larger and more complex spatial systems. A prototype-based process carried out in this way allows for quick and precise testing and customization. The presented method is characterized by the ability to quickly respond to emerging problems in the city, such as heat islands, and the inability to introduce permanent green urban tissue. The main aspect of this solution is the possibility of the application of a digital selection of plants at the early design stage. Due to the implementation of tools that allow for automatic design, the proposed method may be applied by local stakeholders. It does not require specialist knowledge in the field of botany and computational design. This would, however, require creating a library of plant parameters for particular geographical locations according to specific plants’ functionalities as an easily accessible open-source database. Developed for the purpose of this study, the research framework library can be freely enlarged, both in terms of additional parameters and other plant species.

The method of plant selection also allows for easy integration with digital methods of rapid prototyping and the BIM-based design process. This would allow for modeling and evaluating the chosen data through analysis. For instance, the thermal radiation of public spaces could determine where the easy-to-fabricate microclimate prototype structures should be located [64]. Research on the applicability of the proposed solution to reduce the negative effects of urban heat islands is of utmost importance given that, according to the European Climate and Health observatory, heat-related mortality has almost doubled in the last 20 years [65,66].

The automated selection process may be focused in future research on the potential of vegetation to accumulate water or improve soil quality by the absorption of heavy metals [67]. For further investigations, the new parameters could be added to the plants’ cards. This could include plants’ ability to deposit and absorb particulate matter [68]. Since mitigating urban heat islands depends on particular plant qualities, such as Leaf Area Index (LAI), this property could also be included in the library of parameters [20]. The proposed method is easy to apply in building interiors, where the main indicators for health and comfort include indoor air quality as well as thermal and humid conditions [69].

The great advantage of the method is the ability to create independent structures that can be easily located in problematic places and then moved, changed, or expanded. The method allows for the appropriate selection of plant species for a particular site. It also allows for verification in terms of selected parameters, depending on the purpose of implementation. For instance, in particularly sensitive locations, such as healthcare or daycare centers, the plants could be selected based on their unique health-promoting properties such as their ability to reduce anxiety [70]. As Muahram et. al. proves, even exposition to various spectrums of green colors mitigates stress [22] but closer insights into molecular-scale modeling reveal much more; for instance, how human health may benefit from the therapeutic properties of particular types of vegetation on the cellular level [16]. The proposed microclimate structure can be adapted to different types of designs and functions, such as public transport stops or shading canopies. The solution may also include a number of other requirements, such as maintenance demands or limitations.

## 5. Conclusions

The proposition of the method developed for the small-scale green structure prototype provides an important insight into the challenges that go along with the incorporation of vegetation into the design process. In the era of climate change characterized by rising temperatures, expansions of cities, shrinking areas of urban greenery, and demographic changes such as the aging population, the experimental prototype can become a possible rapid solution that could increase the quality of life in cities and bring benefits to human health. The creation of the experimental prototype made it possible to verify the effectiveness and efficiency of the vegetation selection method at the early stages of the project. Moreover, it is highly important to raise the designers’ awareness of the need and opportunities concerning the incorporation of greenery design in the early stages of the design processes, not only for the small-scale prototypes but also buildings and public spaces.

Additionally, the proposed computer method would allow for a more precise and knowledge-based holistic design of greenery. The achieved solution would require consultation with a landscape architect, gardener, or horticulture therapist. However, it would constitute a good basis, supported by hard data, for the development of the concept at later stages of design. The later stages may focus on the search for additional species for enhancing biodiversity in flora and fauna, stimulating olfactory sensations, or increasing effectiveness for air pollutant reduction, depending on the needs. The proposed method could become a response to the often over-complicated interface or redundant data provided at various stages of the work. The plant cards created for the system could consist of selected greenery parameters (SGP), considered during computer analyzes, but could also include an additional full greenery parameters table (FGP) allowing for additional verification of special cases. In the authors’ opinion, creating cards and their variants is the greatest challenge, and an attempt to parametrize individual requirements may be a great research task. However, the advances in this field of research could contribute to creating more healthy urban environments and therefore benefit public health.

## Figures and Tables

**Figure 1 ijerph-20-01198-f001:**
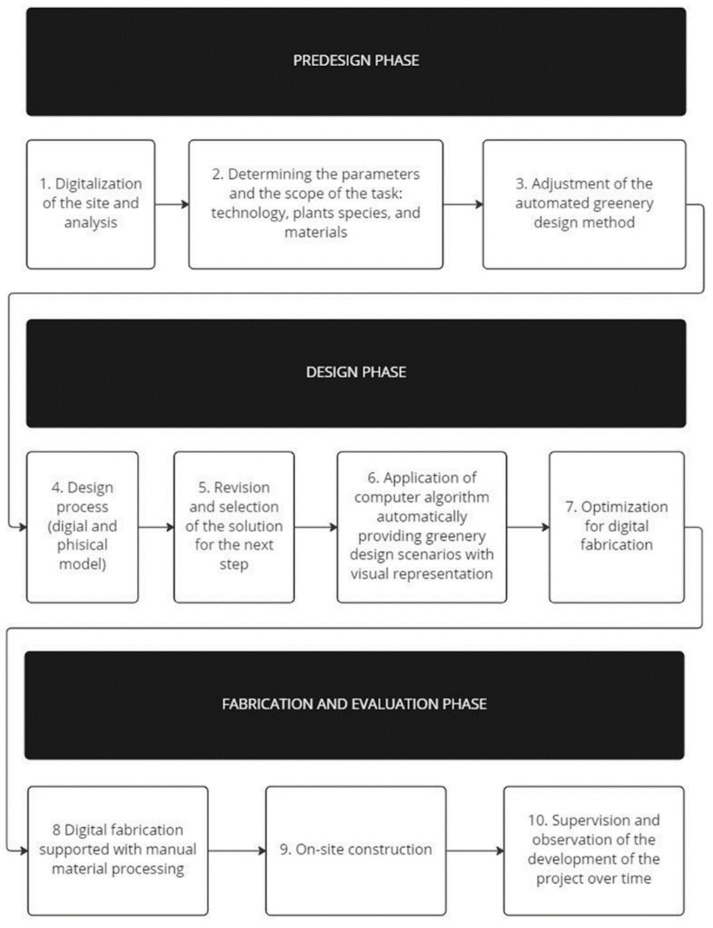
Method diagram.

**Figure 2 ijerph-20-01198-f002:**
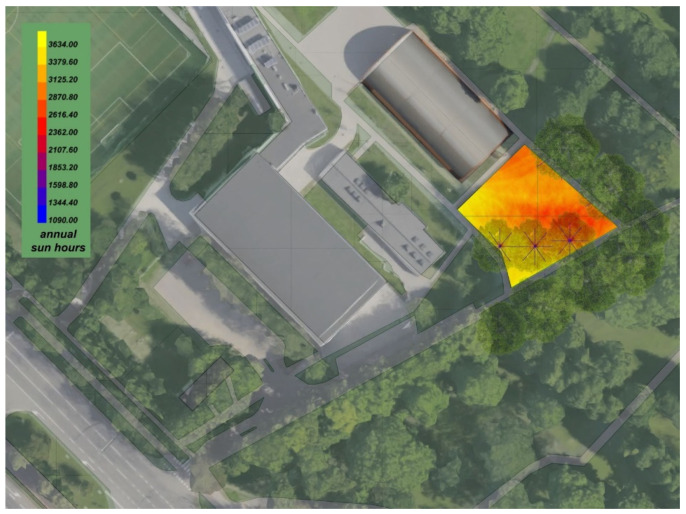
Location of sample scenario.

**Figure 3 ijerph-20-01198-f003:**
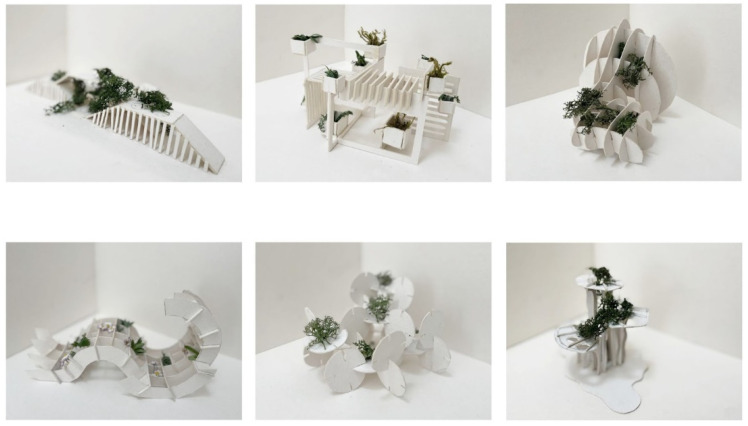
Propositions of the design solution.

**Figure 4 ijerph-20-01198-f004:**
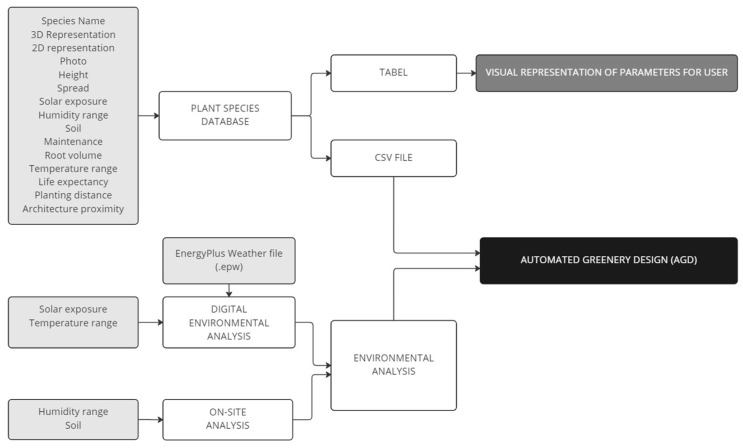
Automated greenery design (AGD) approach.

**Figure 5 ijerph-20-01198-f005:**
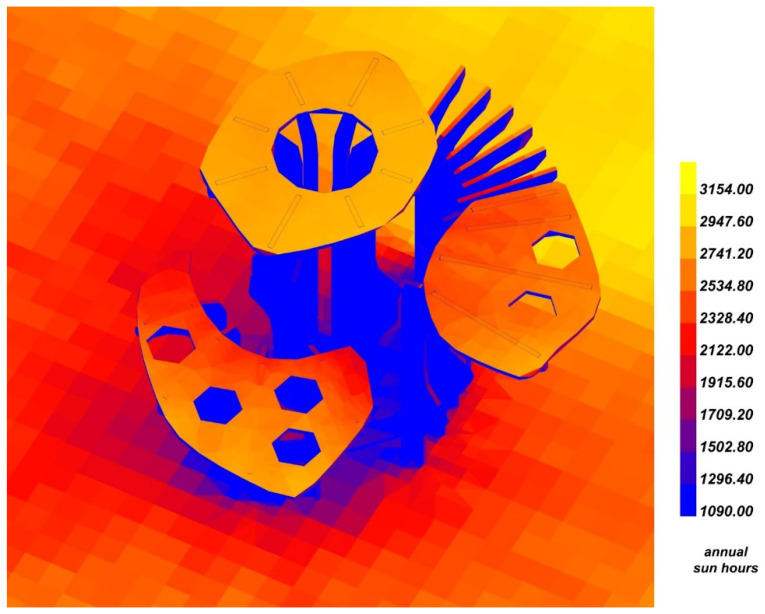
Sun hours analysis of the chosen design.

**Figure 6 ijerph-20-01198-f006:**
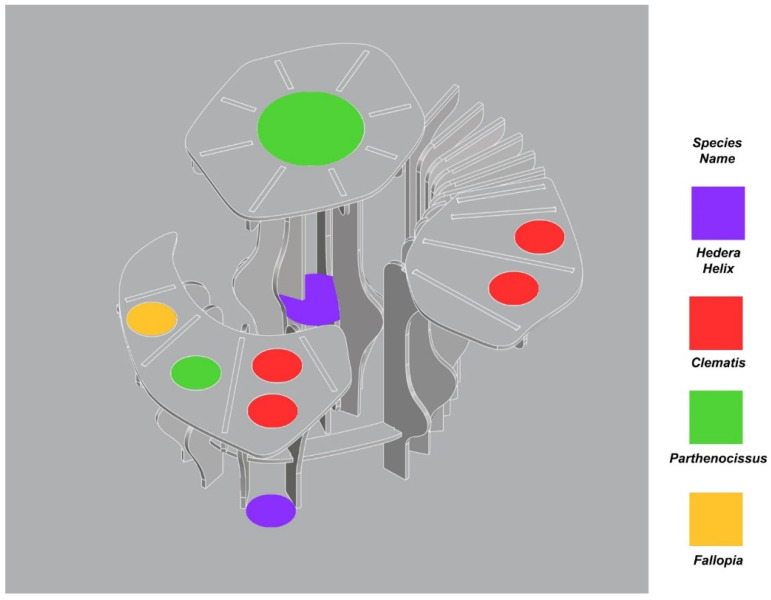
Plant species distribution schematic view.

**Figure 7 ijerph-20-01198-f007:**
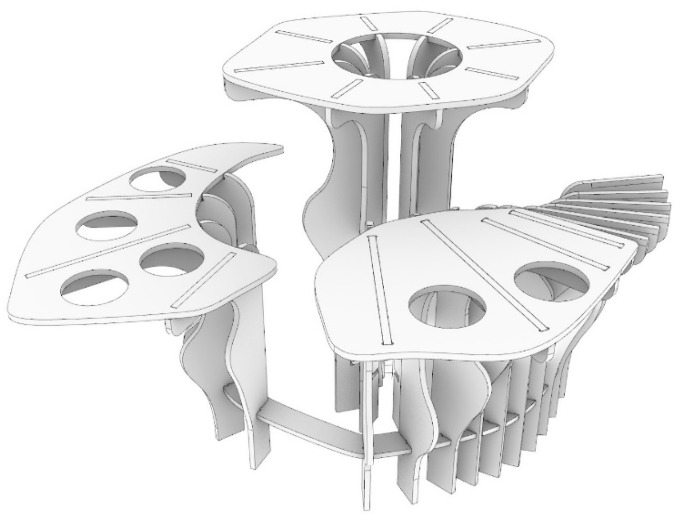
Revised 3D model.

**Figure 8 ijerph-20-01198-f008:**
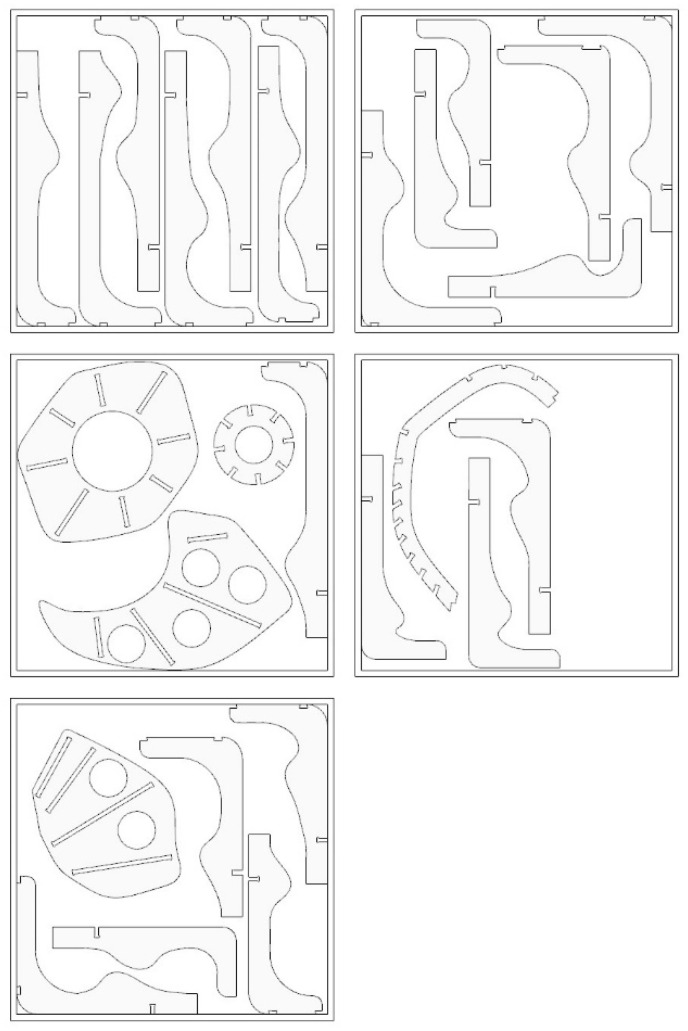
Nesting of the structural elements.

**Figure 9 ijerph-20-01198-f009:**
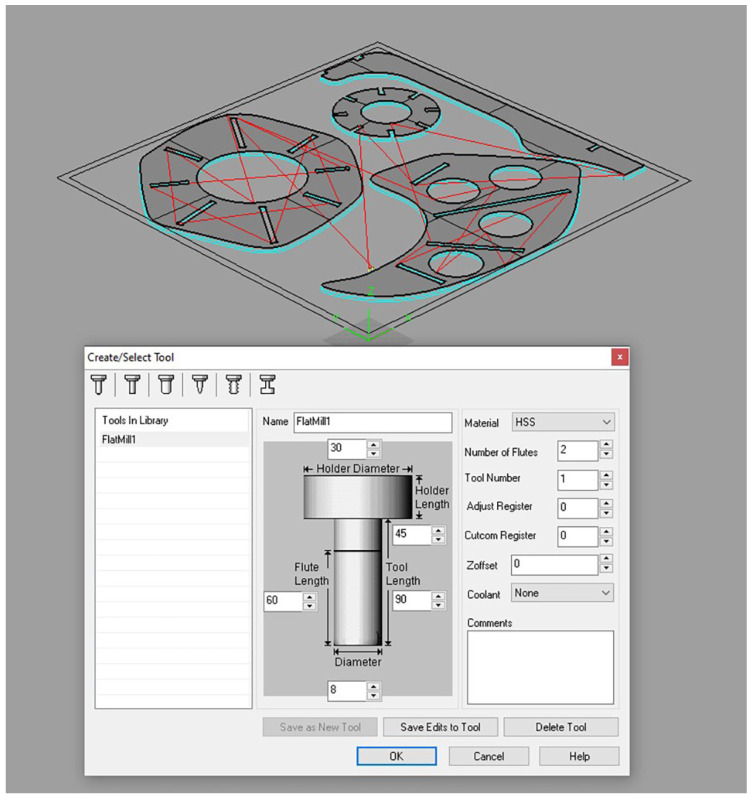
Tool path generation.

**Figure 10 ijerph-20-01198-f010:**
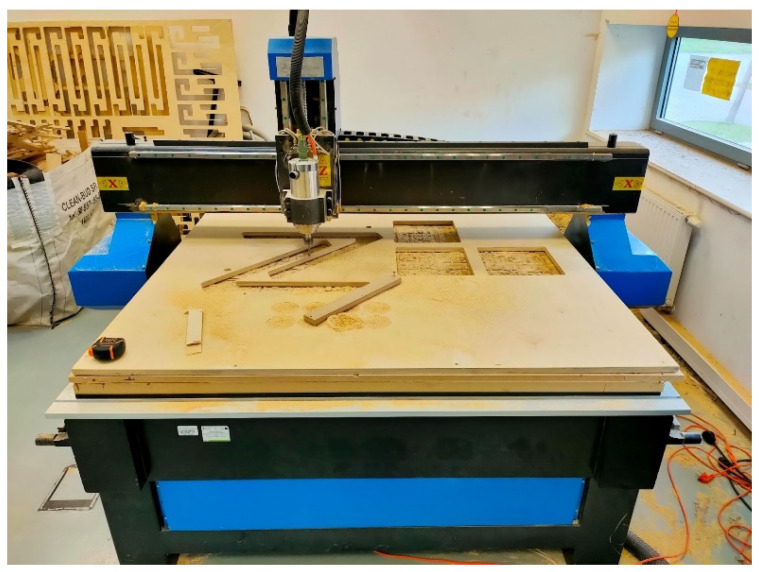
CNC milling.

**Figure 11 ijerph-20-01198-f011:**
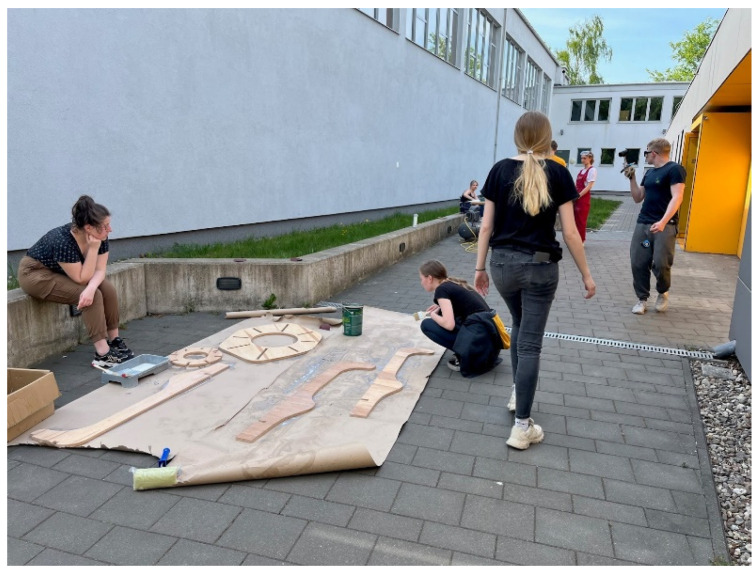
Manual processing.

**Figure 12 ijerph-20-01198-f012:**
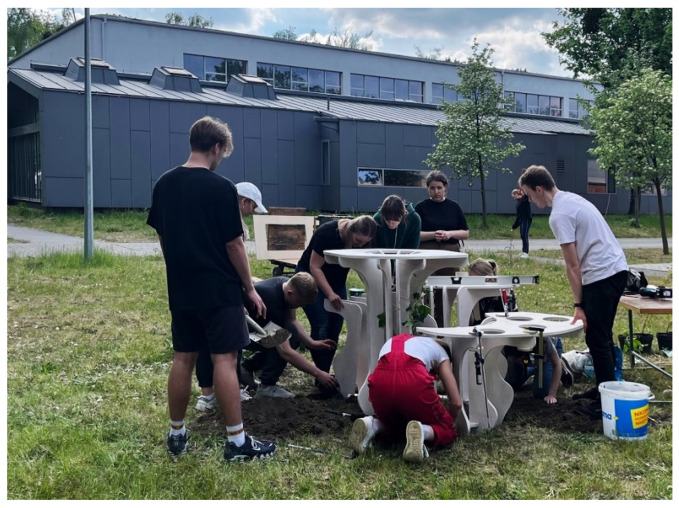
Assembly of the structure.

**Figure 13 ijerph-20-01198-f013:**
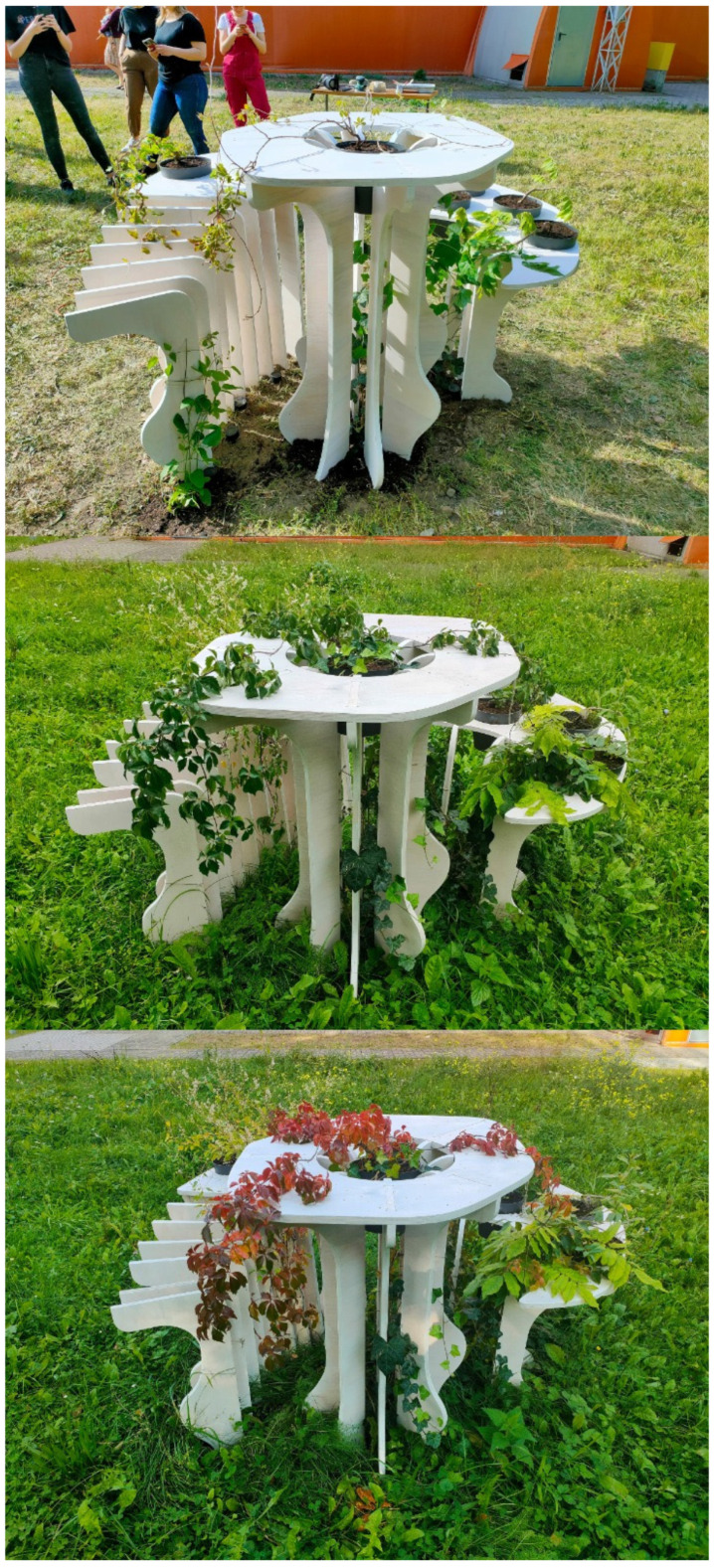
Growth of plants through the seasons (spring, summer, and fall).

**Table 1 ijerph-20-01198-t001:** Selected greenery parameters (SGP) for *Parthenocissus*.

Number	Name	Nursery	Grown	Type of Value	Unit
1.	Category	climber	climber	text	-
2.	Species Name	*Parthenocissus*	*Parthenocissus*	text	-
3.	3D Representation	3D model	3D model	brep	-
4.	2D representation	Green dot	Green dot	spline/point	-
5.	Photo	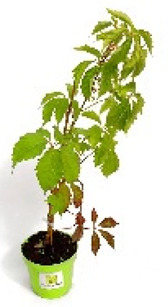	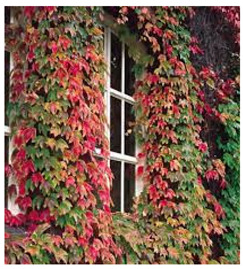	image	-
6.	height	50 cm–150 cm	<15,000 cm	number domain	cm/inch
7.	spread	10 cm–30 cm	<8000 cm	number domain	cm/inch
8.	Solar exposure	1400–4500	1400–4500	number domain	Sun hours
9.	Humidity range	40–80%	40–80%	number domain	%
10.	Soil reaction	pH 6.0–pH 7.5	pH 6.0–pH 7.5	number domain	pH
11.	Maintenance	2	1	number	integer
12.	Root volume	0.3 m^3^	5 m^3^	number	Cubic meters
13.	Temperature range	(−25)–(+ 35)	(−25)–(+ 35)	number domain	C/F
14.	Life expectancy	500 years	500 years	number domain	integer
15.	Planting distance	150 cm	400 cm	number	cm/inch
16.	Architecture proximity	0 cm–60 cm	0 cm–350 cm	number domain	cm/inch

## Data Availability

Not applicable.

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
