# Peer review of "Computer-Aided Greenery Design—Prototype Green Structure Improving Human Health in Urban Ecosystem"

_ijerph, 2023, doi:10.3390/ijerph20021198_

Round 1
Reviewer 1 Report
Dear Authors,
Your article is interesting, however it requires some improvements.
1. The introduction should precisely define the purpose of the research and the approach.
2. It is said in the abstract that ,,The article proposes an automated greenery design method combined with rapid prototyping for such interventions”.
However, the process of automated greenery design should be better described.
It is mentioned about using Rhino plug-ins like Ladybug ang Grasshoper and ,,multiple parameters”, but it should be developed.
What parameters are used in the visual scripts? What optimization criteria were used, whether single-criteria or multi-criteria optimization were applied?
3. The choice of the best solution was poorly justified and is not preceded by any analysis results and optimization results.
4. The mentioned ,,The automated selection process” should be better described.
Best regards
Author Response
Dear Reviewer 1,
We would like to thank the Editor and Reviewers for all the valuable remarks. After a careful analysis of all the comments, we have made the necessary corrections. We hope that the current text will respond to all the remarks.
Could be improved:
Does the introduction provide sufficient background and include all relevant references?
We have explained more clearly the purpose of the study and the research approach we took in the added paragraph:
Thus, the purpose of this research is to propose a new innovative method to develop easy-to-fabricate green structures that can be inserted into dense urban environments contributing to public health and environmental benefits. The proposed approach is based on linking rapid prototyping with BIM tools and automated greenery selection. The presented study develops a new complex digital method of selection and design of greenery based on a parameter spreadsheet and environmental data. Parametrization of the chosen group of plants presented in this paper provides an initial step toward creating a database of plant species that can be used for algorithmic plant selection and distribution in further studies.
Are all the cited references relevant to the research?
We have added additional references and reviewed the previously selected ones:
- Gianoli E. The behavioural ecology of climbing plants. AoB Plants. 2015 Feb 12;7:plv013. https://doi.org/10.1093/aobpla/plv013
- Daniel S. Falster; Mark Westoby, Plant height and evolutionary games, Trends in Ecology & Evolution, VOLUME 18, ISSUE 7, P337-343, JULY 01, 2003, https://doi.org/10.1016/S0169-5347(03)00061-2
- Anna-Maria Llorens; Michelle R. Leishman, Climbing strategies determine light availability for both vines and associated structural hosts, Australian Journal of Botany 56(6) 527-534, 2008, https://doi.org/10.1071/BT07019
- FRANCIS E. PUTZ; HAROLD A. MOONEY, THE BIOLOGY OF VINES, Cambridge University Press, 1991
- Caroline Boisset, The Plant Growth Planner: Two Hundred Illustrated Charts for Shrubs, Trees, Climbers, and Perennials, Prentice-Hall, 1992
Must be improved:
Is the research design appropriate?
The added sections and figures in the article helped to present the research design in a clearer way. Articulating the purpose of the work more explicitly and filling in the gaps in the explanation of the methods and in the presentation of the results improved the design of the research.
Are the methods adequately described?
To describe the method in a clearer way we have decided to add a paragraph regarding the AGD method in section 2.1 Methods:
The AGD method takes into account parameters such as sun exposure, temperature range, moisture, maintenance, soil type, and soil reaction. The analysis of these parameters allowed for the automated selection of specific plants according to the location and characteristics of the created architectural form.
To emphasize the applied automated greenery design method we have decided to add a block diagram as Figure 4. Automated greenery design (AGD) approach.
Are the results clearly presented?
We have cleared the theoretical structure and strengthened the hypothesis and research questions to present the research better. Moreover, we have decided to emphasize on the automated greenery design method in the added 3.2 paragraph the automated selection process. The added section is supported with additional figures. Figure 4. Automated greenery design (AGD) approach – where the AGD method is presented in block-schematic and Figure 6. Plant species distribution schematic view – where final greenery distribution is presented. Moreover, we have decided to edit Figure 5. Sun hours analysis of the chosen design for better results presentation
General
1) The introduction should precisely define the purpose of the research and the approach. It is said in the abstract that: ,,The article proposes an automated greenery design method combined with rapid prototyping for such interventions”.
We have explained more clearly the purpose of the study and the research approach we took in the added paragraph:
Thus, the purpose of this research is to propose a new innovative method to develop easy-to-fabricate green structures that can be inserted into dense urban environments contributing to public health and environmental benefits. The proposed approach is based on linking rapid prototyping with BIM tools and automated greenery selection. The presented study develops a new complex digital method of selection and design of greenery based on a parameter spreadsheet and environmental data. Parametrization of the chosen group of plants presented in this paper provides an initial step toward creating a database of plant species that can be used for algorithmic plant selection and distribution in further studies.
2) However, the process of automated greenery design should be better described. It is mentioned about using Rhino plug-ins like Ladybug ang Grasshopper and ,,multiple parameters”, but it should be developed. What parameters are used in the visual scripts? What optimization criteria were used, whether single-criteria or multi-criteria optimization were applied?
4) The mentioned ,,The automated selection process” should be better described.
For a better understanding of the final results we have added section:
3.2 The automated selection process
The automated greenery design (AGD) is a method for including plants in early-stage design processes, developed at the Digital Technologies and Materials of the Future Laboratory of the GdaÅ„sk University of Technology. The AGD (Figure 4) allows for automated plant selection based on plants’ species parameters combined with site environmental analysis. Parameters that are taken into account can be adjusted according to the designers' needs and characteristics of the project. In the presented experimental model the following parameters were taken into consideration: sun exposure, temperature range, moisture, maintenance, soil type, and soil reaction.
To provide the parameters to the algorithm a database was developed. The database has two forms of representation. The first one consisted of a series of parameters in the form of a spreadsheet in .csv file format, that was applied to the algorithm. The second one was developed in the form of a table for presenting the parameters to the designer and user (Table 1). The environmental evaluation was based on on-site and digital analysis. The soil type and reaction were measured at the selected location. The humidity range, temperature range, and sun exposure (Figure 5) were calculated in the process of digital analysis on the basis of environmental data from the EnergyPlus Weather file (.epw) located in the Nowy Port district.
The automation was conducted with Rhinoceros®, Grasshopper®, and Ladybug® tools. The outcome of the process was the optimal scenario for plant selection computed for the chosen design of the structure. (Figure 6) In the computational process, the environmental analysis was combined with plant parameters that allowed for the solution presentation in the form of a schematic view that points out a specific plant location. Subsequently, this scenario was applied in a prototype experimental model.
The added section is supported with additional figures. Figure 4. Automated greenery design (AGD) approach – where the AGD method is presented in block-schematic and Figure 6. Plant species distribution schematic view – where final greenery distribution is presented. Moreover, we have decided to edit Figure 5. Sun hours analysis of the chosen design for better results presentation.
3) The choice of the best solution was poorly justified and is not preceded by any analysis results and optimization results.
We have decided to justify of best solution selection in the added paragraph:
All of the projects provided acceptable solutions to design problems. The concept selected for the construction, however, distinguished itself from the others with the quality of achieved design goals. Firstly, the chosen proposal presented the most diverse distribution of greenery positions on the structure relative to different levels of sun exposure. This allowed for the automatic plant selection algorithm to use every plant species from the provided list when matching the designated spots with different kinds of vegetation. Secondly, the chosen design, when compared to the other proposals, offered the most efficient material usage. It used only 5 sheets of plywood, whereas most of the other designs used exceedingly more. Thirdly, the selected structure was the easiest to build. It used the smallest number of timber joints out of all of the proposals and also, in contrast with the other designs, provided specifically designed holes for the pots, therefore ensuring the easiest construction and fast greenery planting.
Reviewer 2 Report
This is a well considered and interesting paper that aims to provide relatively short-term and inexpensive methods for reducing the urban heat island effect and adding small areas of green space to urban habitats.
For publication the following should be addressed:
1) Editors should identify and/or fix the (relatively minor) language problems to improve clarity for readers in some places. Here is an example of sentences needing editing to improve clarity:
From Darvin’s studies [45] to modern-day architectural experimentation on climbing plants and their effects on microclimate are sufficiently documented [46]. Not only, this group of plants is a good fit when designing for a better wellbeing in cities, but they are also applicable when it comes to a particular health benefits to the environment like removing air pollution [47] or improving cooling performance of buildings [48].
A general check for errors is needed, e.g. to catch problems like 'XXXXX' in line 253.
2) Figure 2 is unclear. I'm assuming the rectangles are buildings, but why does one have a highlight? What do the stars indicate? The tall red canes? Also, the location is unclear from the figure because the landmarks mentioned in the text are not shown in the figure (e.g. 6 Alley Road).
3) Trademark symbols should be used for proprietary software packages, and plant genera should be italicized.
4) Table 1 does not make sense. Please clarify it. E.g., why are there symbols for 2nd and 3rd representations but not first? Sun hours per what? Where did the values for (for example) height and spread come from, what do the maintenance rankings mean, and why is pH listed only as < 7.5??
5) For the section on 'Material selection,' please clarify exactly what you're writing about at the beginning. Material for what?
6) The discussion does well at summarizing the findings of this paper and presenting potential broader effects and future directions.
Author Response
Dear Reviewer 2,
We would like to thank the Editor and Reviewers for all the valuable remarks. After a careful analysis of all the comments, we have made the necessary corrections. We hope that the current text will respond to all the remarks.
Could be improved:
Are the methods adequately described?
To describe the method more precisely we have decided to add a paragraph regarding the AGD method in section 2.1 Methods:
The AGD method takes into account parameters such as sun exposure, temperature range, moisture, maintenance, soil type, and soil reaction. The analysis of these parameters allowed for the automated selection of specific plants according to the location and characteristics of the created architectural form.
To emphasize the applied automated greenery design method we have decided to add a block diagram in Figure 4. Automated greenery design (AGD) approach.
Are the results clearly presented?
For a better understanding of the final results, we have decided to add Figure 6. Plant species distribution schematic view and edit Figure 5. Sun hours analysis of the chosen design. Moreover, we have added section: 3.2 The automated selection process:
The automated greenery design (AGD) is a method for including plants in early-stage design processes, developed at the Digital Technologies and Materials of the Future Laboratory of the GdaÅ„sk University of Technology. The AGD (Figure 4) allows for automated plant selection based on plants’ species parameters combined with site environmental analysis. Parameters that are taken into account can be adjusted according to the designers' needs and characteristics of the project. In the presented experimental model the following parameters were taken into consideration: sun exposure, temperature range, moisture, maintenance, soil type, and soil reaction.
To provide the parameters for the algorithm a database was developed. The database has two forms of representation. The first one consisted of a series of parameters in the form of a spreadsheet in .csv file format, that was applied to the algorithm. The second one was developed in the form of a table for presenting the parameters to the designer and user (Table 1). The environmental evaluation was based on on-site and digital analysis. The soil type and reaction were measured at the selected location. The humidity range, temperature range, and sun exposure (Figure 5) were calculated in the process of digital analysis on the basis of environmental data from the EnergyPlus Weather file (.epw) located in the Nowy Port district.
The automation was conducted with Rhinoceros®, Grasshopper®, and Ladybug® tools. The outcome of the process was the optimal scenario for plant selection computed for the chosen design of the structure. (Figure 6) In the computational process, the environmental analysis was combined with plant parameters that allowed for the solution presented in the form of a schematic view that points out a specific plant location. Subsequently, this scenario was applied in a prototype experimental model.
General
1) Editors should identify and/or fix the (relatively minor) language problems to improve clarity for readers in some places. Here is an example of sentences needing editing to improve clarity:
From Darvin’s studies [45] to modern-day architectural experimentation on climbing plants and their effects on microclimate are sufficiently documented [46]. Not only, this group of plants is a good fit when designing for a better wellbeing in cities, but they are also applicable when it comes to a particular health benefits to the environment like removing air pollution [47] or improving cooling performance of buildings [48].
A general check for errors is needed, e.g. to catch problems like 'XXXXX' in line 253.
We have reviewed the text and proofread the current version to avoid such situations. Here is the corrected example:
Climbing vegetation growth has gained considerable research interest, from Darvin’s studies [45] to modern-day experimentations [46]. This group of plants is a good fit when designing for better well-being in cities. They are applicable in regard to particular health and environmental benefits such as removing air pollution [47] or improving the cooling performance of buildings [48].
2) Figure 2 is unclear. I'm assuming the rectangles are buildings, but why does one have a highlight? What do the stars indicate? The tall red canes? Also, the location is unclear from the figure because the landmarks mentioned in the text are not shown in the figure (e.g. 6 Alley Road).
We have decided to edit Figure 2 to present the context in a clear and easy-to-read way.
3) Trademark symbols should be used for proprietary software packages, and plant genera should be italicized.
We have reviewed the text and added trademark symbols and italicized plant species names.
4) Table 1 does not make sense. Please clarify it. E.g., why are there symbols for 2nd and 3rd representations but not first? Sun hours per what? Where did the values for (for example) height and spread come from, what do the maintenance rankings mean, and why is pH listed only as < 7.5??
We have decided to edit the table accordingly to the remarks. We would like to mention that 2d and 3d representations referred to 3-dimensional and 2-dimensional representation to avoid confusion we have decided to mark them as 2D and 3D Representation. Sun hours are presented as per year. pH levels were changed from the open number domain to the closed number domain. The table is a part of the broader material that comes along with the additional explanations. Height, spread, maintenance, and pH level parameters come from the sources of the data that were published in several research studies regarding vegetation listed below:
- Gianoli E. The behavioural ecology of climbing plants. AoB Plants. 2015 Feb 12;7:plv013. https://doi.org/10.1093/aobpla/plv013
- Daniel S. Falster; Mark Westoby, Plant height and evolutionary games, Trends in Ecology & Evolution, VOLUME 18, ISSUE 7, P337-343, JULY 01, 2003, https://doi.org/10.1016/S0169-5347(03)00061-2
- Anna-Maria Llorens; Michelle R. Leishman, Climbing strategies determine light availability for both vines and associated structural hosts, Australian Journal of Botany 56(6) 527-534, 2008, https://doi.org/10.1071/BT07019
- FRANCIS E. PUTZ; HAROLD A. MOONEY, THE BIOLOGY OF VINES, Cambridge University Press, 1991
- Caroline Boisset, The Plant Growth Planner: Two Hundred Illustrated Charts for Shrubs, Trees, Climbers, and Perennials, Prentice-Hall, 1992
To present the data more clearly, the source of the parameters got an additional explanation in the changed paragraphs:
Climbing vegetation growth has gained considerable research interest, from Darvin’s studies [45] to modern-day experimentations [46]. This group of plants is a good fit when designing for better well-being in cities. They are applicable in regard to particular health and environmental benefits such as removing air pollution [47] or improving the cooling performance of buildings [48].
The research on planting preferences and characteristics of growth for particular species provided parameters that were used for the Automated Greenery Design method (AGD). The sources of the data were several research studies regarding vegetation [49-54], in particular, the characteristics of plant growth were based on “The Plant Growth Planner” by Caroline Boisset [53] and preferred sun exposure was developed according to “Begrünte Architektur” by Rudi Baumann [54]. The final data is the outcome of the evaluation and selection of available plant parameters. The research aimed to analyze and provide the set of parameters that affect plant vegetation in relation to the architectural environment. The data used in the research was created by the Laboratory of Digital Technologies and Materials of the Future, Faculty of Architecture, GdaÅ„sk University of Technology. Parameterization of plants’ data conducted in the laboratory is part of a broader research program on the possibility of automation of design processes. An example of the gathered and curated data is presented in the table below (Table 1).
The group of plant species from the climbers family was chosen from an available database produced by the Laboratory of Digital Technologies and Materials of the Future. The most relevant criterion for this experiment was to provide a diversified spectrum of solar exposure so that any area on and around the planned installation can be matched with a particular species for planting. The chosen species are:
- Clematis
- Fallopia
- Hedera helix
- Parthenocissus
In subsequent steps of the study, plants’ parameters in combination with climatic data allowed for the automatic analysis process to be carried out as an integrated element of spatial structure design.
5) For the section on 'Material selection,' please clarify exactly what you're writing about at the beginning. Material for what?
We have decided to change the title of section 2.2.3 to Experimental model construction: material and fabrication technology selection, moreover to clarify the purpose of the section we have added the paragraph:
For the strategy of building and testing the prototype to work, adequate consideration of construction material was necessary. The main drivers behind the choice of materials for the construction of a prototype green structure were determined by the project's aim to implement the presence of greenery support in urban scenarios with minimum effort and maximum adjustment to the existing environment.
Round 2
Reviewer 1 Report
Dear Authors,
after improvements, the paper can be accepted in a present form.
best regards